# Water Chlorophyll a Estimation Using UAV-Based Multispectral Data and Machine Learning

**Xiyong Zhao** [1,2,3], **Yanzhou Li** [1], **Yongli Chen** [2], **Xi Qiao** [1,3,*] and **Wanqiang Qian** [3,*]

1 College of Mechanical Engineering, Guangxi University, Nanning 530004, China
2 Guangxi Bossco Environmental Protection Technology Co., Ltd., Nanning 530007, China
3 Shenzhen Branch, Guangdong Laboratory for Lingnan Modern Agriculture, Genome Analysis Laboratory of the Ministry of Agriculture and Rural Affairs, Agricultural Genomics Institute at Shenzhen, Chinese Academy of Agricultural Sciences, Shenzhen 518120, China
* Correspondence: qiaoxi@caas.cn (X.Q.); qianwanqiang@caas.cn (W.Q.)

**Abstract:** Chlorophyll a (chl-a) concentration is an important parameter for evaluating the degree of water eutrophication. Monitoring it accurately through remote sensing is thus of great significance for early warnings of water eutrophication, and the inversion of water quality from UAV images has attracted more and more attention. In this study, a regression method to estimate chl-a was proposed; it used a small multispectral UAV to collect data and took the vegetation indices as intermediate variables. For this purpose, ten monitoring points were selected in Erhai Lake, China, and two months of monitoring and data collection were conducted during a cyanobacterial bloom period. Finally, 155 sets of valid data were obtained. The imaging data were obtained using a multispectral UAV, water samples were collected from the lake, and the chl-a concentration was obtained in the laboratory. Then, the images were preprocessed to extract the information from different wavebands. The univariate regression of each vegetation index and the regression using band information were used for comparative analysis. Four machine learning algorithms were used to build the model: support vector machine (SVM), random forest (RF), extreme learning machine (ELM), and convolutional neural network (CNN). The results showed that the effect of estimating the chl-a concentration via multiple regression using vegetation indices was generally better than that via regression with a single vegetation index and original band information. The CNN model obtained the best results ($R^2$ = 0.7917, RMSE = 8.7660, and MRE = 0.2461). This study showed the reliability of using multiple regression based on vegetation indices to estimate the chl-a of surface water.

**Keywords:** chl-a; multiple regression; UAV; vegetation index; machine learning





## 1. Introduction

Since eutrophication began to appear in inland lakes in the 1930s, approximately 40–50% of lakes and reservoirs in the world have been affected by eutrophication to varying degrees, and lake eutrophication has become one of the most intractable water environmental problems [1]. Chlorophyll a (chl-a) concentration is an important indicator to measure the abundance of phytoplankton in lake water and an important parameter to determine water quality, biological status, and eutrophication degree [2,3]. Chl-a, which is a photosynthetic pigment found in algae species [4], has been considered the largest weighted factor to calculate the Carlson trophic state indices [5]. Assessments of the timing and extent of algal biomass and eutrophication states in aquatic ecosystems may frequently be performed using extensive long-term chl-a concentration measurements [6]. These eutrophication insights can help us to understand primary production, biogeochemical cycling, and overall inland water quality, which can result in environmental changes and useful mitigation tactics [7,8].

Using remote sensing technology to measure a series of parameters indicating water quality (temperature, transparency, turbidity, photosynthetic pigment concentration,

colored dissolved organic matter, etc.), it was shown that this variability in aquatic ecosystems can be effectively captured [9,10]. Data sources are mainly multispectral sensors carried by satellites, such as Moderate Resolution Imaging Spectroradiometer (MODIS) Aqua and Terra, Medium Resolution Imaging Spectrometer (MERIS), Operational Land Imager (OLI), and Sentinel 2-A/B [11–19]. However, low spatial and temporal resolution, atmospheric correction challenges, cloud cover, slow data–product turn-around times, and the cost of some satellite products can limit the application of satellite remote sensors for monitoring water quality in inland systems [20]. In addition, the concentration of chl-a at the water surface varies at different times of the same day. It increases with the temperature and is usually highest at noon. This also leads to space-time errors between water sample data and satellite data during a study [21]. Unmanned aerial systems (UASs) (remotely piloted remote sensing platforms, i.e., UAVs), show great potential for bridging the gap between in situ water sampling and satellite remote sensing. UASs collect high-resolution aerial data with minimal atmospheric disturbance from cloud coverage, allow flexible flight planning with rapid turn-around times, and are available in a variety of wavelength combinations [22]. There are already many commercial UAVs equipped with multiband and multispectral sensors, their performance is stable, and they can meet daily monitoring needs.

In traditional chl-a remote sensing monitoring research, according to the different reflectivities of chl-a in different spectral bands, some band algorithms for chl-a were proposed, such as the single-band threshold, band interpolation algorithms, and band ratio algorithms [23,24], which are simple and suitable for general estimation. The three-band method [25,26] and four-band method [27] weaken the influence of turbid water on chl-a and improve the accuracy of the model. However, the band algorithm requires specific band data; therefore, it requires special multispectral or hyperspectral sensors. It is widely used in satellite remote sensing monitoring, and large UAVs are required to carry hyperspectral equipment in UAV monitoring.

With the development of artificial intelligence, machine learning has become an important technology in remote sensing image processing [28–30], and has also been widely used in water quality monitoring [31]. In many chl-a remote sensing monitoring studies, Peterson et al. [32,33] used an extreme learning machine (ELM) to estimate chl-a and obtained good results. An ELM also showed good performance in organic carbon estimation [34]. Philipp M et al. used a convolutional neural network (CNN) to invert chl-a and obtained promising results, which were better than those from the Bayesian regularization (BR) algorithm model [35,36]. Many scholars have used support vector machines (SVMs) and random forests (RFs) to estimate chl-a, and all of them achieved good results [37–40]. The effect is better than that of the models established using artificial neural networks (ANNs) [41,42], dynamic programming (DP) [43], generalized linear models (GLMs) [44], and other algorithms. Therefore, we chose SVM, RF, ELM, and CNN to be evaluated.

At present, the estimation model of chl-a is more established by using regression of the band reflectivity [45–47]. In the monitoring of chl-a using a small multispectral UAV, multiband multispectral sensors mainly include the red (R), green (G), blue (B), red-edge (RE), and near-infrared (NIR) bands, which can be used to calculate the vegetation indices to represent the concentration of chl-a. Based on G. Edna's correlation analysis on different vegetation indices and chl-a [48], this study proposed a method to estimate chl-a concentration by using vegetation indices as an intermediate variable, which provided a new idea for small multispectral UAV monitoring of chl-a.

In many studies of chl-a remote sensing inversion, many of them analyzed image data through existing algorithms, but only used a small amount of data as validation, and lacked a lot of actual data to support their claims [27,49]. Most of the regression algorithms used to predict chl-a adopt the method of multiple sampling points in one experiment to collect data, which has a small amount of data and is in the same environment [42,50]. The lack of

long-term monitoring and data collection makes the performance of the model worse in complex environments.

In light of this context, this study aimed to monitor the chl-a concentration and multi-spectral data of Erhai Lake (Yunnan, China) for two months. The main objectives of this study were as follows: (1) explore the feasibility of using vegetation indices as intermediate variables to estimate the concentration of chl-a; (2) the effect of using vegetation indices as an intermediate variable to estimate chl-a concentration was analyzed compared with that of using single vegetation index and original variable; (3) analyze the applicability of SVM, RF, ELM, and CNN machine learning algorithms in the estimation of chl-a concentration.

## 2. Materials and Methods

### 2.1. Study Area

The data collection area was located in the northwestern part of Erhai Lake, as shown in Figure 1. Erhai Lake is located in Dali city, Yunnan Province, China (25°36′–25°58′ N, 100°06′–100°18′ E, 1972 m above sea level). It has an average depth of 10.5 m and a surface area of about 251 km². Since the 1990s, with the development of the social economy, the disturbance from human activities has been increasingly aggravating Erhai Lake, its water quality has deteriorated, a large number of aquatic plants have disappeared, and the eutrophication of water has become gradually obvious, which has gradually degraded the ecological environment of Erhai Lake [51]. The water quality of Erhai Lake is mainly classified as class II in the dry season and class III in the rainy season, and large blooms mainly occur around October.

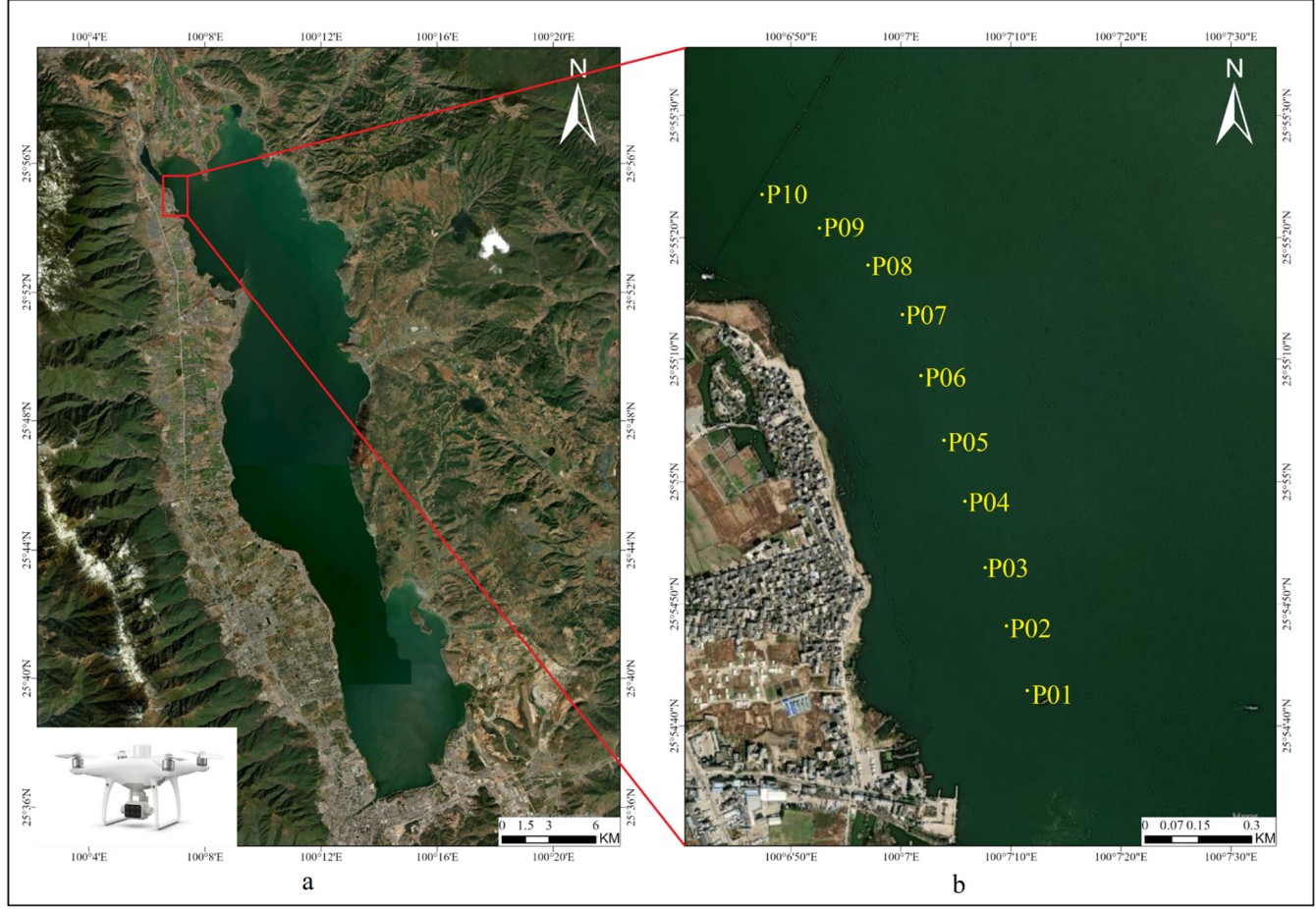

**Figure 1.** (**a**) Study area overview and its location in Erhai Lake, with the used unmanned aerial vehicle in the inset, and (**b**) locations of the sampling points.

The concentration of chl-a in the northern coastal area of Erhai Lake was high in previous years. However, the shore water is shallow, the grass is more abundant, and the bottom color has a greater impact on measurements. Therefore, 100 m away from the shore, one sampling point every 120 m and a total of 10 sampling and monitoring points were selected.

### 2.2. Data Acquisition and Preprocessing

Multispectral image acquisition and lake water samples were collected from 10 monitoring sites over a total of 60 days between September and November 2021. Experiments were conducted every three days. Due to rain and other weather-related reasons, a field experiment was not possible, and images were affected by surface ships and wind; therefore, the data were not sufficiently accurate. Therefore, a total of 18 experiments were conducted and 155 groups of data were collected.

The Phantom 4 Multispectral (P4M) UAV produced by Dajiang Innovation Technology Co., Ltd. (DJI), Shenzhen, China, was used in this study. DJI P4M and its multispectral camera were used for image collection in this experiment. The multispectral sensor has a six-camera array, including a high-definition color sensor and five monochrome sensors: R, G, B, RE, and NIR. It has a light intensity sensor at the top that captures solar irradiance and records it in image files. When the data is post-processed, the solar irradiance data can be used for illumination compensation of the image to eliminate the interference of ambient light on data acquisition; no whiteboard calibration was required. The specifications of DJI P4M are shown in Table 1.

**Table 1.** DJI P4M UAV parameters.

| | | | |
|---|---|---|---|
| **Aircraft** | **Type** | | Quadcopter |
| | **Position accuracy** | | Vertical: $\pm$ 0.5 m; horizontal: $\pm$ 1.5 m |
| | **Take-off weight** | | 1484 g |
| | **Time of flight** | | About 27 min |
| | **Wind loading rating** | | 10 m/s |
| | **Image position compensation** | | Relative positions are compensated in photo EXIF coordinates |
| **Image sensor** | **Visible light channel** | | Pixels: 2 million 80 thousand |
| | **Multispectral sensor:** all sensors have 2 megapixels | **G channel** | Spectral band: 450 $\pm$ 16 nm |
| | | **B channel** | Spectral band: 560 $\pm$ 16 nm |
| | | **R channel** | Spectral band: 650 $\pm$ 16 nm |
| | | **RE channel** | Spectral band: 730 $\pm$ 16 nm |
| | | **NIR channel** | Spectral band: 840 $\pm$ 26 nm |

Data from DJI website; http://www.dji.com/cn/p4-multispectral/specs (accessed on 30 September 2022).

The flight path was delimited on DJI GS Pro software to pass through each monitoring point. The flight altitude was 10 m, the camera angle was $-90°$, and the exposure was automatic. The resolution of the photos obtained was $1600 \times 1300$, with an accuracy of approximately 0.53 cm/pixel.

Each experiment began at 1 p.m. when temperatures were the highest and the chl-a concentrations were the highest throughout the day. First, the UAV flew according to the predetermined route to obtain multispectral images. Then, water sample collection was carried out. Water samples were collected at a depth of 20 cm from the surface of the water using a water sampler. Finally, the water quality parameters (chl-a concentration, turbidity, and water temperature) of the water samples were tested in the laboratory.

For the water samples, we used the chl-a concentration module of the HX-200 multi-parameter controller produced by Beijing Hongxinhengce Technology Co., Ltd. (Beijing, China) to analyze the chl-a concentration. It makes use of the characteristics of chl-a with absorption peaks and reflection peaks in the spectrum. The monochromatic light at the absorption peak band of the chl-a spectrum was emitted into the water. The chl-a in the water absorbs the energy of the light and reflects monochromatic light with another

wavelength. The light intensity reflected by chl-a is directly proportional to the content of chl-a in the water, which is used to calculate the concentration of chl-a.

To collect higher-resolution images, the UAV was flown at a lower altitude and the surface of the lake would ripple under the influence of the wind, causing the image to be affected by the reflection of light. Therefore, the main processing of the image was the extraction and removal of reflective areas. Finally, the data from non-reflective areas were determined to reduce the influence of reflection on the results.

The fuzzy C-means (FCM) algorithm is a common clustering algorithm based on the fuzzy theory introduced by J. C. Birdek [52]. It maximizes the similarity of data objects within classes and minimizes the similarity of data objects between classes. It is an improvement on the traditional C-means clustering. It divides the samples according to the degree of the image pixels belonging to the clustering center and has been widely used in image segmentation. The reflective area extraction effect of the FCM algorithm is shown in Figure 2.

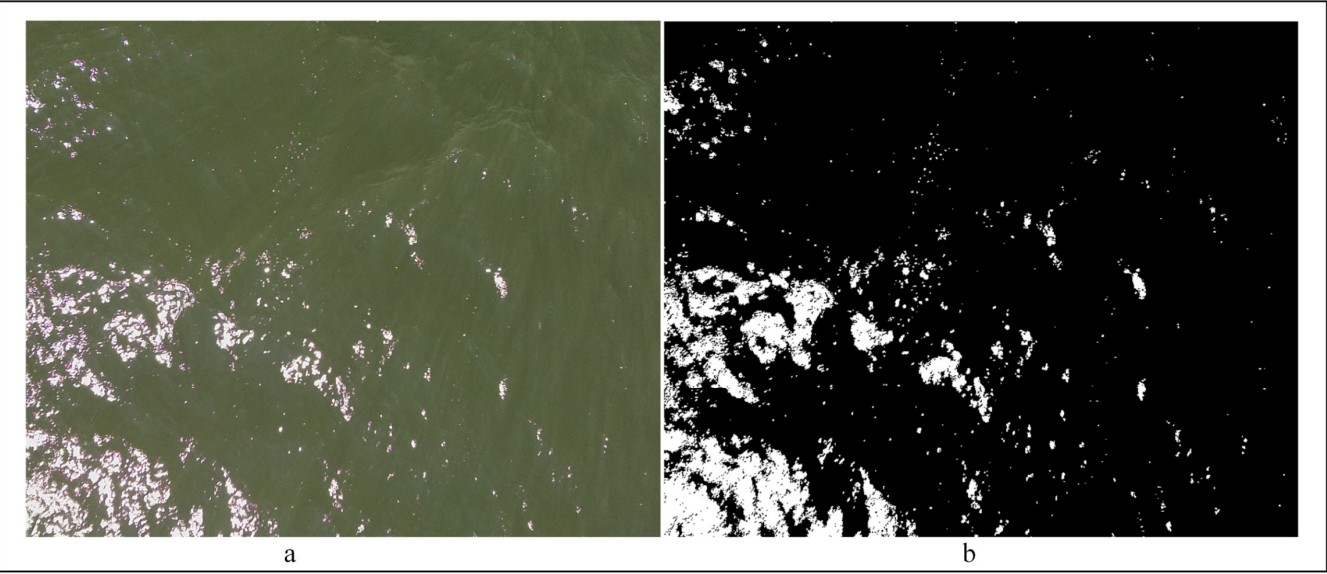

**Figure 2.** Extraction of the reflective area of the image: (**a**) original visible light image and (**b**) reflection area extraction effect image.

After the extraction of the reflective region in the visible image, the non-reflective region was covered on the multispectral band image using a mask, and the non-reflective region of the multispectral band image was extracted. Then, the gray value of each multispectral band image was extracted and all gray values of each image were averaged. Finally, the reflectance of different bands was calculated according to the gray value.

### 2.3. Vegetation Index

According to the vegetation indices (band ratio algorithm) containing the five bands R, G, B, RE, and NIR in relevant studies, 28 common vegetation indices with formulas were finally selected for this research, among which 14 contained only the RGB visible band and 14 contained the RE or NIR band. The vegetation indices and their formulas are shown in Table 2.

**Table 2.** Vegetation indices and their formulas and value ranges.

| Vegetation Index | Formula | Value Range |
|---|---|---|
| Normalized difference vegetation index [49] | $NDVI = (NIR - R)/(NIR + R)$ | $[-1,1]$ |
| Normalized difference vegetation structure index [53] | $NDVSI = (NIR - (R + G)/2)/(NIR + (R + G)/2)$ | $[-1,1]$ |
| Normalized green–blue difference index [54] | $NGBDI = (G - B)/(G + B)$ | $[-1,1]$ |
| Normalized green–red difference index [54] | $NGRDI = (G - R)/(G + R)$ | $[-1,1]$ |
| Ratio normalized difference vegetation index [55] | $RNDVI = ((NIR - R)/(NIR + R)) * (NIR/R)$ | $(-\infty, +\infty)$ |
| Ratio vegetation index [56] | $RVI = NIR/R$ | $(0, +\infty)$ |
| Red–green–blue vegetation index [57] | $RGBVI = (G^2 - R * B)/(G^2 + R * B)$ | $[-1,1]$ |
| Red-edge chlorophyll index [58] | $CIREDEDGE = (NIR/RE) - 1$ | $[-1, +\infty)$ |
| Red–green ratio index [54] | $RGRI = R/G$ | $(0, +\infty)$ |
| Visible atmospherically resistant index [59] | $VARICREEN = G - R/G + R - B$ | $(-\infty, +\infty)$ |
| Visible band difference vegetation index [54] | $VDVI = (2 * G - R - B)/(2 * G + R + B)$ | $[-1,1]$ |
| Vegetative [60] | $VEG = G/(R^a * B^{1-a}) (a = 0.667)$ | $(-\infty, +\infty)$ |
| Blue–green ratio index [61] | $BGRI = B/G$ | $(-\infty, +\infty)$ |
| ENGBDI [61] | $ENGBDI = (G^2 - B^2)/(G^2 + B^2)$ | $[-1,1]$ |
| Two-band enhanced vegetation index [62] | $EVI2 = (2.5 * (NIR - R))/(NIR + 2.4 * R + 1)$ | $[-0.71, 1.25]$ |
| Blue normalized vegetation index [63] | $BNDVI = (NIR - B)/(NIR + B)$ | $[-1,1]$ |
| Color index of vegetation extraction [64] | $CIVE = 0.441 * R - 0.881 * G + 0.385 * B + 18.787$ | $[17.906, 19.586]$ |
| Difference vegetation index [65] | $DVI = NIR - R$ | $[-1,1]$ |
| Enhanced normalized difference vegetation index [66] | $ENDVI = (NIR + G - 2 * B)/(NIR + G + 2 * B)$ | $[-1,1]$ |
| Excess green index [54] | $EXG = 2 * G - R - B$ | $[-2,2]$ |
| Excess green minus excess red [54] | $EXGR = EXG - 1.4 * R - G$ | $[-4.4,2]$ |
| Green chlorophyll index [58] | $CIGREEN = (NIR/G) - 1$ | $[-1, +\infty)$ |
| Green normalized difference vegetation index [67] | $GNDVI = (NIR - G)/(NIR + G)$ | $[-1,1]$ |
| Green–red ratio index [68] | $GRRI = G/R$ | $(0, +\infty)$ |
| KIVU [69] | $KIVU = (B - R)/G$ | $(-\infty, +\infty)$ |
| Modified simple ratio red edge [70] | $MSRRE = ((NIR/RE) - 1)/((NIR/RE) + 1)$ | $(-1,1)$ |
| Modified single ratio [67] | $MSR = ((NIR/R) - 1)/((NIR/R) + 1)$ | $(-1,1)$ |
| Normalized difference red-edge index [71] | $NDRE = (NIR - RE)/(NIR + RE)$ | $[-1,1]$ |

B, blue; G, green; NIR, near-infrared; R, red; RE, red edge. Data from the IDB website; https://www.indexdatabase.de/ (accessed on 25 August 2022).

### 2.4. Modeling Techniques

The algorithms used for image preprocessing and regression in this study were all completed in MATLAB software, version 2019a. The parameters in the following models were the results after the gradient test.

### 2.4.1. Support Vector Machine (SVM)

An SVM is a kind of generalized linear classifier that classifies bivariate data according to supervised learning. Its decision boundary is a hyperplane with a maximum margin to solve the learning samples, and the problem can be transformed into a convex quadratic programming problem [72]. Compared with logistic regression and neural networks, support vector machines provide a clearer and more powerful way to learn complex nonlinear equations. The optimization problem of an SVM considers both empirical risk and structural risk minimization; therefore, it is stable. An SVM is mainly used for classification and regression analysis. The main parameters of the SVM model in this study were as follows: SVM type(-s), 3 (epsilon-SVR); kernel type(-t), 2 (radial basis function); cost(-c), 2.2; gamma(-g), 2.8; epsilon(-p), 0.01. The main parameters of the vegetation index univariate regression, original variable multiple regression, and vegetation indices multiple regression models remained the same.

### 2.4.2. Random Forest (RF)

In 2001, Leo Breiman combined classification trees into an RF, that is, the use of variables and data were randomized to obtain a certain number of classification trees. Then, the results of the classification trees were summarized, and the random forest algorithm was proposed [73]. An RF is a type of ensemble learning. Ensemble learning is a very

popular machine learning strategy, and almost all problems can be improved by using its ideas, and it is more stable when analyzing small sample data. The basic starting point is to gather algorithms and various strategies together. Ensemble learning can be used for both classification problems and regression problems, and it is often seen in the field of machine learning. The main parameters of the RF model in this study were as follows: leaf, 5; tree, 500. The main parameters of the vegetation index univariate regression, original variable multiple regression, and vegetation indices multiple regression models remained the same.

### 2.4.3. Extreme Learning Machine (ELM)

An ELM is a kind of machine learning system or method based on a feedforward neural network (FNN), which is suitable for both supervised and unsupervised learning problems [74]. The most significant feature of an ELM is that the input weights and the bias of hidden nodes are randomly generated within a given range, which was shown to have high learning efficiency and strong generalization ability. The main purpose of training is to solve the weights of the input layer. An ELM is widely used in classification, regression, clustering, feature learning, and other problems [75]. The main parameters of the ELM model in this study were as follows: number of hidden neurons (N), 30; transfer function (TF), sig (sigmoidal function); type, 0 (regression). The main parameters of the vegetation index univariate regression, original variable multiple regression, and vegetation indices multiple regression models remained the same.

### 2.4.4. Convolutional Neural Networks (CNN)

A CNN is a class of feedforward neural networks with a deep structure and convolutional computation, which is one of the representative algorithms of deep learning [76]. A CNN imitates biological visual perception mechanism construction, which can carry out supervised learning and unsupervised learning. The convolution kernel parameter shares in the hidden layers and the sparsity of the connections between layers allow the CNN to use grid-like topological features with less computation. The main parameters of the CNN model in this study were as follows. In the univariate regression of the vegetation index, only the final regression layer was used. The multiple regression model included one convolution layer, one pooling layer, and two fully connected layers. Convolution layer: convolution kernel size, $3 \times 1$; step length, 16; podding, same. Pooling layer: pool kernel size, $2 \times 1$; step length, 2. The number of neurons in the two fully connected layers: both were 9 in the original multivariate regression model; in the multivariate regression model of vegetation indices, both were 52. Training parameters: epoch, 50; batch size, 4; learn rate, 0.005. The training parameters of all CNN models were the same.

### 2.5. Modeling Strategy and Validation Metrics

In this study, the performance of the model was evaluated using the coefficient of determination ($R^2$) [56], root-mean-square error (RMSE) [57], and mean relative error (MRE) [51], where the related formulas are as follows:

$$R^2 = 1 - \frac{\sum_{i=1}^{n}(\hat{y}_i - y_i)^2}{\sum_{i=1}^{n}(y_i - \overline{y})^2} \tag{1}$$

$$RMSE = \sqrt{\frac{1}{n}\sum_{i=1}^{n}(\hat{y}_i - y_i)^2} \tag{2}$$

$$MRE = \frac{\sum_{i=1}^{n}\frac{\hat{y}_i - y_i}{y_i}}{n} \tag{3}$$

where $y_i$ is the measured value, $\hat{y}_i$ is the predicted value, $\overline{y}$ is the average of the measured values, and $n$ denotes the number of samples.

The larger the $R^2$ and the smaller the RMSE and MRE, the better the model performance.

*2.6. Data Analysis*

The maximum, minimum, median, mean, and standard deviation of the observed chl-a concentration data were analyzed statistically. Furthermore, the significance analysis of the estimated concentration of chl-a was carried out. The software used was IBM SPSS Statistics (v. 25.0).

## 3. Results

*3.1. Descriptive Statistics*

To build and test the model, the total dataset was randomly divided into a training set and a test set. In this study, the Rand perm function was used for random classification. The original ratio was 3:1, but the number cannot contain decimals, and considering the convenience of calculation and statistics, the number of training sets was 120, and the rest were test sets. The statistical values of the training and test datasets for the chl-a concentration estimation are shown in Table 3. The datasets were analyzed using the maximum, minimum, median, mean, and standard deviation. According to the data in Table 3, the distribution of the whole dataset and training and testing datasets was similar. Therefore, the training set and test set could be used to establish the model and test the model accuracy.

**Table 3.** Descriptive statistics of the whole dataset, training dataset, and test dataset.

| Dataset | Sample Size | Maximum (μg/L) | Minimum (μg/L) | Median (μg/L) | Mean (μg/L) | Standard Deviation (μg/L) |
|---|---|---|---|---|---|---|
| Whole dataset | 155 | 84.74 | 8.03 | 19.26 | 24.89 | 15.12 |
| Training dataset | 120 | 81.32 | 8.03 | 18.67 | 23.63 | 13.68 |
| Test dataset | 35 | 84.74 | 11.75 | 23.62 | 29.23 | 18.65 |

*3.2. Univariate Regression Results of Vegetation Indices*

Table 4 shows the test results of the univariate regression model established using the four algorithms on 28 vegetation indices. According to the analysis of the best results of each vegetation index among the four algorithms, there were 10 vegetation indices with $R^2$ values greater than 0.5 (9 of which contained NIR bands), 6 of which were between 0.3 and 0.5, and 12 of which were less than 0.3. The DVI achieved the best results using the ELM model ($R^2$ = 0.6923). In the univariate regression of the vegetation index, the ELM achieved the best results in most vegetation index regressions, especially when the $R^2$ values were greater than 0.5. The second was the RF, which performed better than the CNN and SVM in most cases.

**Table 4.** Coefficient of determination *($R^2$)* values of vegetation index univariate regression.

| Vegetation Index | SVR | RF | ELM | CNN |
|---|---|---|---|---|
| NDVI | 0.4649 ** | 0.5760 ** | 0.4783 ** | 0.4246 ** |
| NDVSI | 0.3799 * | 0.4865 ** | 0.5660 ** | 0.3701 * |
| NGBDI | 0.0571 | 0.3313 * | 0.2453 * | 0.0612 |
| NGRDI | 0.0005 | 0.4952 ** | 0.2711 * | 0.0002 |
| RNDVI | 0.6063 ** | 0.5762 ** | 0.5742 ** | 0.6111 ** |
| RVI | 0.5863 ** | 0.5746 ** | 0.4573 ** | 0.5080 ** |
| RGBVI | 0.0354 | 0.0312 | 0.0123 | 0.0348 |
| CIREDEDGE | 0.0468 | 0.0062 | 0.0088 | 0.0408 |
| RGRI | 0.0038 | 0.4723 ** | 0.1863 | 0.0013 |
| VARIGREEN | 0.0142 | 0.1536 | 0.2286 * | 0.0008 |
| VDVI | 0.0356 | 0.0073 | 0.0029 | 0.0355 |
| VEG | 0.0172 | 0.0396 | 0.1450 | 0.0161 |

**Table 4.** *Cont.*

| Vegetation Index | SVR | RF | ELM | CNN |
|---|---|---|---|---|
| BGRI | 0.0577 | 0.3520 * | 0.4116 * | 0.0732 |
| ENGBDI | 0.0490 | 0.3525 * | 0.3511 * | 0.0625 |
| EVI2 | 0.4439 ** | 0.5758 ** | 0.5200 ** | 0.4395 * |
| BNDVI | 0.2685 ** | 0.2907 * | 0.2730 * | 0.2663 * |
| CVIE | 0.0842 | 0.0086 | 0.0003 | 0.0842 |
| DVI | 0.4282 * | 0.6057 ** | 0.6923 ** | 0.4249 * |
| ENDVI | 0.1299 | 0.1461 | 0.1071 | 0.1297 |
| EXG | 0.0682 | 0.0001 | 0.0001 | 0.0682 |
| EXGR | 0.5798 ** | 0.6030 ** | 0.6794 ** | 0.5800 ** |
| CIGREEN | 0.3376 * | 0.3194 * | 0.6213 ** | 0.3516 * |
| GNDVI | 0.2982 * | 0.3310 * | 0.6235 ** | 0.3128 * |
| GRRI | 0.0030 | 0.4883 ** | 0.1844 | 0.0001 |
| KIVU | 0.0470 | 0.0506 | 0.2461 * | 0.0465 |
| MSRRE | 0.0435 | 0.0058 | 0.0048 | 0.0418 |
| MSR | 0.4649 ** | 0.5832 ** | 0.4835 ** | 0.4246 ** |
| NDRE | 0.0435 | 0.0052 | 0.0048 | 0.0418 |

Levels of significance: * $p$-value $< 0.05$; ** $p < 0.01$.

### 3.3. Original Band Reflectance Regression Results

We used data from the five original bands as independent variables for regression as a reference, and the results are shown in Table 5. The results showed that the ELM model achieved the best results ($R^2 = 0.7202$, RMSE = 8.3023, and MRE = 0.2728), and in terms of the $R^2$, RMSE, and MRE values, the results were significantly better than the models established by other algorithms.

**Table 5.** Original band reflectance regression results.

| Algorithm | $R^2$ | RMSE (µg/L) | MRE |
|---|---|---|---|
| SVM | 0.6753 ** | 9.0535 | 0.2907 |
| RF | 0.6426 ** | 10.0772 | 0.3084 |
| ELM | 0.7202 ** | 8.3023 | 0.2728 |
| CNN | 0.6598 ** | 9.6963 | 0.3040 |

Level of significance: ** $p < 0.01$.

### 3.4. Multiple Regression Results of the Vegetation Indices

First, linear correlation analysis was conducted between the vegetation indices and the chl-a concentration (represented by $R^2$), and the analysis results are shown in Figure 3. The figure shows that the variable with the highest correlation was the RNDVI, and its $R^2$ value was 0.3435. The RNDVI was originally developed to differentiate between vegetation and soil to estimate crop growth without the need to measure near-infrared wavelengths, and it is not typically included in UAS aquatic ecosystem monitoring studies. There were 12 variables with correlations greater than 0.1, among which 11 vegetation index formulas required NIR band data. Moreover, the 10 vegetation indices with $R^2$ values greater than 0.5 in the univariate regression of the vegetation indices were all in the above 12 vegetation indices.

To explore the difference between the effects of multiple regression with the vegetation indices as the independent variable and univariate regression with the vegetation index, 28 planting indices were used as the independent variables; the multiple regression results of the model that was established using the four regression algorithms are shown in Table 6. The results showed that the CNN model achieved the best results ($R^2 = 0.7917$, RMSE = 8.7660, and MRE = 0.2461), followed by the ELM model, RF model, and SVM model. The results of the evaluation indices $R^2$, RMSE, and MRE values, were consistent with the above conclusions.

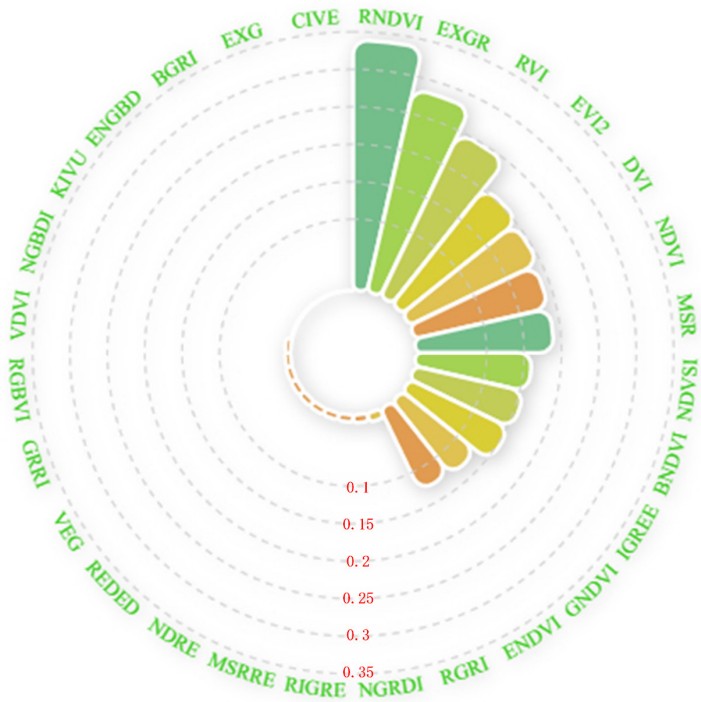

**Figure 3.** Results of the linear correlation analysis ($R^2$).

**Table 6.** Results of the multiple regression of the vegetation indices.

| Algorithm | $R^2$ | RMSE (μg/L) | MRE |
|---|---|---|---|
| SVM | 0.5800 ** | 12.4570 | 0.2894 |
| RF | 0.7086 ** | 10.3713 | 0.2891 |
| ELM | 0.7810 ** | 9.5050 | 0.2576 |
| CNN | 0.7917 ** | 8.7660 | 0.2461 |

Level of significance: ** $p < 0.01$.

The results of the multiple regression test data of the vegetation indices are shown in Figure 4. As seen from the figure, the dispersion degree increased with decreasing $R^2$ values, the CNN model was the most stable, and the dispersion degree was the lowest. Because there were fewer data with high concentrations, the prediction results of the model with high concentrations were not as good as those with low concentrations. In particular, the SVM model had poor performance when predicting high concentrations.

Finally, we used the combination of vegetation indices and original band data as independent variables for regression, and the results are shown in Table 7. The best performance was from the CNN model ($R^2 = 0.7710$, RMSE = 9.1317, and MRE = 0.2485), but its performance was inferior to that of the vegetation index multiple regression model.

**Table 7.** Multivariate regression results of the vegetation index and original band information.

| Algorithm | $R^2$ | RMSE (μg/L) | MRE |
|---|---|---|---|
| SVM | 0.5489 ** | 12.8715 | 0.2963 |
| RF | 0.7065 ** | 10.3394 | 0.2904 |
| ELM | 0.7299 ** | 9.9185 | 0.2751 |
| CNN | 0.7710 ** | 9.1317 | 0.2485 |

Levels of significance: ** $p < 0.01$.

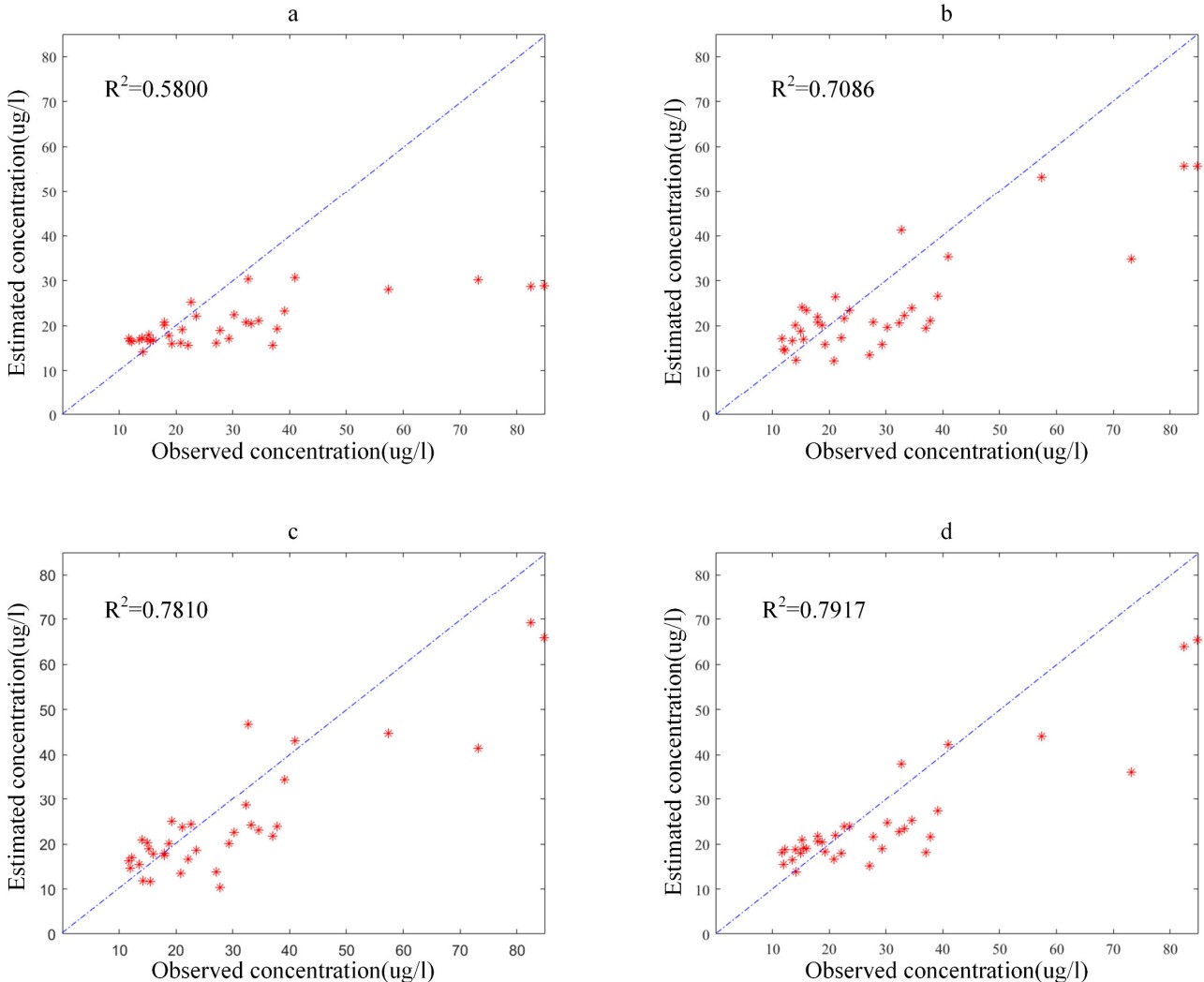

**Figure 4.** Test results of multiple regression models of the vegetation indices: (**a**) SVM model, (**b**) RF model, (**c**) ELM model, and (**d**) CNN model.

In general, compared with the univariate regression of the vegetation index and the original band information regression, the effect of the multiple regression of the vegetation indices was significantly improved. Among the different algorithms, ELM displayed the best performance in the univariate regression of the vegetation index using band information regression, and the CNN displayed the best performance in the multiple vegetation indices variable regression. The CNN multivariate model based on the vegetation indices achieved the best results among all the models ($R^2$ = 0.7917, RMSE = 8.7660, and MRE = 0.2461).

## 4. Discussion

### 4.1. Correlation between Vegetation Index and Chl-a

Many multiband multispectral sensors are specifically designed to record the peak reflectance and absorption characteristics of chl-a. The band of the multispectral sensor of the DJI P4M UAV used in this study has some similarities with the characteristic band of chl-a. Studies using UAVs and satellites to monitor water quality also found that vegetation indices including red and near-infrared bands, such as the RVI and NDVI, perform well when estimating chl-a [77,78]. In the univariate regression of the vegetation index in this study, the ELM model of the vegetation index DVI achieved the best result ($R^2$ = 0.6923), and the regression results of RVI and NVDI were also greater than 0.5. The DVI is mainly used to

detect vegetation growth status and vegetation coverage; the NDVI is the most widespread vegetation index and it can reflect the background influences of the plant canopy, such as soil and dead leaves [79]; and the RVI is very sensitive to vegetation coverage. Moreover, there were 10 vegetation indices with $R^2$ values greater than 0.5, of which 9 contained the NIR band and 7 contained the NIR and R bands. This was also consistent with the results that water with a higher chl-a concentration is significantly different from ordinary water in the NIR band [80]. The importance of the NIR band in the results also suggested that UAVs equipped with multispectral sensors have advantages in chl-a estimation over UAVs equipped with ordinary HD cameras. In this study, the DVI achieved the best results, but there were slight differences in the best-realized vegetation index among different studies, which indicated that the vegetation indices could produce different results when using data under different circumstances; therefore, it was not rigorous enough to use a single vegetation index to represent the concentration of chl-a.

### 4.2. Improvement of the Chl-a Estimation Effect Using Multiple Regression of Vegetation Indices

In the multiple regression of the vegetation indices, the CNN model achieved the best result ($R^2$ = 07917, RMSE = 8.7660, and MRE = 0.2461). Compared with the wave segment reflectance, the effect of multiple regression was significantly improved. In addition, among the models established by the four algorithms, only the accuracy of the SVM model decreased ($R^2$ decreased by 0.0073), while the accuracy of the other algorithm models increased, among which the $R^2$ values of the RF increased by 0.0660 and the $R^2$ values of the ELM increased by 0.0608. In particular, the effect of the CNN model was significantly improved ($R^2$ values increased by 0.1319). This indicated that using the vegetation indices as an intermediate variable could explore more information in the data and increase the number of features, which was more conducive to the regression model established by the machine learning algorithm, thus improving the chl-a regression effect. The linear correlation analysis results between the vegetation indices and chl-a concentration showed that the $R^2$ value of the RNDVI was the largest ($R^2$ = 0.3435) and the correlation was generally small; therefore, it is not reliable to use a single vegetation index to represent the chl-a concentration. By using vegetation index univariate regression, there was a nonlinear relationship between the chl-a concentration and $R^2$ values that were significantly improved, but the effect of using vegetation indices multiple regression was better. This also showed that a single vegetation index had many features that could not be involved, and the multiple regression of multiple vegetation indices could enrich these features and make the regression estimation more accurate. Using the combination of vegetation indices and original band data as independent variables, the results showed that the performance was lower than that of the multivariate regression of vegetation indices. It showed that this combination produced some redundancy, resulting in poor model performance. The multivariate regression of vegetation indices was the most suitable for the estimation of chl-a concentration with multispectral data.

Compared with satellites and UAVs equipped with hyperspectral sensors, small multispectral UAVs are easy to operate and save labor and material costs. In the estimation of chl-a with hyperspectral data, most of them used the characteristic bands of chl-a for regression [22,38,78,81]. However, since the band of multispectral data is not completely the same as the characteristic band of chl-a, it becomes the best choice to estimate the concentration of chl-a via regression using vegetation indices as the intermediate variable.

### 4.3. Adaptability of the Algorithm

In the results of univariate regression using the vegetation indices and multiple regression using band reflectance, the ELM algorithm outperformed the other algorithms. The ELM has a single hidden layer and is considered better than other shallow learning systems, such as the SVM, in terms of the learning rate and generalization ability [74]. When using the vegetation indices for multiple regression, the CNN model achieved the best results. In univariate regression, since the single variable could not be convolved, only

the regression layer was used in the CNN algorithm, and its effect was worse than that of other algorithms. When using the vegetation indices for multiple regression, 28 vegetation indices were convolved and, finally, regression was performed. It provided better results than the other algorithms. This shows that when the number of features was large, the CNN could better extract effective features and improve the regression results [82]. An SVM and RF have a certain degree of robustness but do not have a deep structure; in the face of complex situations, the effect will be worse, but is suitable for small-sample analysis [83,84]. A CNN has an automatic feature extraction capability, which is more suitable for analysis in complex situations [85].

*4.4. Deficiency and Prospects*

In this study, the vegetation indices were used as an intermediate variable to estimate the concentration of chl-a. After verification, this method was more suitable for small multi-spectral UAVs monitoring the concentration of chl-a. In this study, the concentration of chl-a was mainly below 50 μg/L, and there were few data above 50 μg/L. Therefore, the high-concentration data were slightly insufficient in the model training, which could lead to greater error in the prediction of high concentrations compared with low concentrations. In the future, more data will be added to increase the model performance. In this study, 28 vegetation indices were used for analysis. The use of hyperspectral sensors that have a higher spectral resolution can also be considered for use. In this way, more vegetation indices can be calculated [86] so that the chl-a estimation model displays better performance in complex environments.

**5. Conclusions**

To explore the method of chl-a estimation for small multispectral UAVs, this study proposed a multiple regression method using the vegetation indices as an intermediate variable. Then, the FCM clustering segmentation algorithm was used to extract and remove the reflective area in the image, and then the gray value of other areas of the image was extracted and the mean value was taken. The next step was to calculate the reflectance according to the extracted gray value to calculate 28 vegetation indices. Then, the calculated vegetation index was used as the independent variable and the detected chl-a was used as the dependent variable, and the data set was divided into the training set and the test set using random classification. Finally, the data sets were established and tested using SVM, RF, ELM, and CNN regression algorithms, and $R^2$, RMSE, and MRE were used to evaluate the model performance to identify the model with the best performance. The univariate regression of the vegetation index and the multiple regression of the original band data were used as references. Using the above process, the following findings were obtained in this study:

(1) It was feasible to establish a chl-a regression estimation model with the vegetation indices as an intermediate variable. Moreover, compared with the previous regression estimation using band data and using a single vegetation index to express chl-a concentration, the effect was significantly improved.

(2) Previous studies found that, due to different data and other reasons, when a single vegetation index is used to represent chl-a, the best-performing vegetation index is not completely consistent. The best-performing vegetation index in this study was the DVI, which was different from many studies. Therefore, it is not reliable to use a single vegetation index to represent the chl-a concentration. Only by combining multiple vegetation indices can we find more features in the data and make the results more convincing.

(3) The ELM model displayed the best performance in the univariate regression of the vegetation index and multiple regression of the original band data. In the multiple regression of the vegetation indices, the CNN model obtained the best results. Therefore, in the case of univariate regression and fewer variables, the ELM could analyze

data better and faster. In the case of more variables, the CNN could better utilize the advantages of the convolutional layer and extract more features from the data.

**Author Contributions:** Conceptualization, Y.L.; Methodology, X.Q.; Software, X.Z., X.Q. and W.Q.; Formal analysis, X.Z.; Resources, Y.C.; Data curation, X.Z. and Y.L.; Writing – original draft, X.Z.; Writing – review & editing, X.Z., X.Q. and W.Q.; Project administration, Y.L.; Funding acquisition, Y.L. and Y.C. All authors have read and agreed to the published version of the manuscript.

**Funding:** The work in this study was supported by the National Key R&D Program of China (2021YFD1400100 and 2021YFD1400101) and the Guangxi Ba-Gui Scholars Program of China (2019A33).

**Conflicts of Interest:** The authors declare no conflict of interest.

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
