# Peer review of "Water Chlorophyll a Estimation Using UAV-Based Multispectral Data and Machine Learning"

_drones, doi:10.3390/drones7010002_

Round 1

Reviewer 1 Report (Previous Reviewer 3)

The manuscript addresses the estimation of chlorophyll in water from UAV multispectral data. The authors implemented the recommendations from my previous review.

Some minor improvements:

In the abstract "R2 = 0791", there seems to be a missing comma, present only two decimal places for R2, RMSE, and MRE.

Table 2 in the "calculation formula" collumn double chech de vegetation indices acronyms (e.g. "RVI" instead of "RVII"). Moreover, the authors should include the reference to the studies that providede the vegetation indices equations.

The newly added text in Section 4.4. could be modified, here is a proposal: "The use of hyperspectral sensors that an higher spectral resolution, can also be considered to be used. This way, more vegetation indices can be calculated [67], so that the chl-a estimation model has better performance in complex environments."

Author Response

Reviewer 2 Report (New Reviewer)

Manuscript, entitled Water Chlorophyll a Estimation Using UAV-based Multispec- tral Data „. Research is interesting and well structure with some promising data. The article reports the results show that the effect of estimating chl-a concentration by multiple regression using vegetation indices were generally better than that by regression with a single vegetation index and original band information. The CNN model obtained the best results (R2=0.7917, RMSE=8.7660 and MRE=0.2461). This study shows the reliability of using vegetation indices as an intermediate variable in the estimation of chl-a using a UAV-based multispectral data.

1-      Title should be modified to (Using UAV-based Multispectral Data and Machine Learning Model to estimate Chlorophyll a Concentration of Surface Water in Erhai Lake, China) to reflect all information about the manuscript.

Abstract:

2-      Line 19. Please write how many samples were measured across two months?

3-      Line 22 and line 23. Please remove this sentence (Finally, the vegetation indices were calculated and multiple regression estimation was carried out).

4-      Line 27 and 28. Please remove this sentence (The performance of the models was evaluated by the coefficient of determination (R2), Root Mean-Square Error (RMSE) and Mean Relative Error (MRE)).

5-      Please add some results about the best spectral indices which reflect the good relationships with Chlorophyll a Concentration?

6-      Last sentence should be modified to (This study shows the reliability of using multiple regression based on vegetation indices to estimate of chl-a of surface water in Erhai Lake).

7-      Line 31. Please correct (R2=07917, RMSE=8.7660 and MRE=0.2461) to (R2=0.79, RMSE=8.766 and MRE=0.246).

Introduction:

8-      I recommend to the authors to read this paper because they are close to this work based on ground remote sensing. And it can help them in introduction and discussion.

  •  Assessment of Water Quality in Lake Qaroun Using Ground-Based Remote Sensing Data and Artificial Neural Networks. DOI: 10.3390/w13213094.

9-      What is the novelty (originality) of the work? And what is new in your work that makes a difference in the body of knowledge? What has been done that goes beyond the existing research

10-  How would this research work advance the previous work done in the existing field of study and/or across other fields?

11-  The objectives of this study should be written in clear points

For example, the objective of this study was to; (i) study the performance of published spectral indices to assess the chlorophyll a of surface water  ; (ii) evaluating the performance of machine learning models  based on individual spectral index, spectral bands, combined spectral indices to the chlorophyll a of surface water ; (iii) and elss…………………..

Materials and methods

12-  Please write about the method which used to assess chlorophyll a concentration?

13-  How did you do quality control (QC) and quality assurance (QA) on the obtained data to validate the conclusions?

14-  Please added the references of each spectral index in table 2?

15-  Please remove the equations 1, 2, and 3 and write only the references of them in the text? They are common using in several manuscript.

Results

16-  The statistics analysis should be written in materials and methods for that the section 3.1 Descriptive Statistics should be mention in materials and methods. Before writing more details about the in results.

17-   It is important to add the values of Chla in tables for ten locations in table or figure to show the range and change in chlorophyll a through this period and present these results in the text.

18-   Table 4, 5, 6 and figure 4. Please present the significant of R2 at *, **, *** statistically significant at p 0.05, p 0.01, and p 0.001?.

19-   I suggest to the authors to test the machine learning models based on   combination data of all spectral indices and spectral bands to compare with others.

Discussion

20-  Discussion should be modified according to suggestion above in results.

21-  Discussion need to support by more previous studies

22-  Please, write the practical applications of your work in a separate section, before the conclusions and provide your good perspectives.

Round 2

Reviewer 2 Report (New Reviewer)

1-The title of the manuscript must be modified by adding the machine learning models to reflect all information about the manuscript. I think the half of the manuscript focus on Modeling Techniques.

2- The title of table 2 must be reflected the all information into the table. Please modify it.

3- I know Table 3 is not only a statistical analysis of the data, but also represents the reliability of all the following model inputs. But also you have to mention about statistical analysis of the data in materials and methods. And present only reliability of all the following model inputs in results.

4.  The authors said that significance analysis refers to the correlation analysis between variables and results, while R2 is the digital index of results and cannot be used for significance analysis. But as I know, from the model we can extract the data sets of observed and predicted and then we can test the significant. Please check this point again.  

5. According to Concern # 12: did the authors use the values of spectral indices in the models, or the values of spectral bands of spectral indices in the models. Please confirm it? Anyway, add the best results of them in the manuscript? Other studies found that the combination of spectral indices is better. Then discuss them with other studies.

Author Response

This manuscript is a resubmission of an earlier submission. The following is a list of the peer review reports and author responses from that submission.

Round 1

Reviewer 1 Report

Review of article Estimation of Chlorophyll a in Water Based on a Small Multi- spectral UAV by Zhao et. al. 2022.

The authors of Zhao et al. present a study regarding the mapping of chlorophyll in a lake, using small UAV-based multispectral images, regression, machine learning and ground truth. While the data assembly is impressive and promising, the processing seems reasonable and sound, the manuscript is lacking clear method description, proper citation and a general repeatability of the proposed methods. It remains with the reader to trust the text and tables. Im afraid a rejection is necessary.

UAV imaging is a rising technique with great potential but many caveats, this should be properly addressed.

Additionally, it is recommended good to make research data available for transparency, and the exactly mentioned classification and training purposes, or share the code and algorithms.

Chapter 1.

Many speculative arguments here, the biggest is that for a large water body the use of satellite imagery is obvious. S-2 and Landsat over more bands. The UAV flew 10 m above the water, that’s low and does it capture a reasonable amount of surface area?

Additionally, flying over water is endangering the UAV and risks the whole operation due to unforeseen battery issues, wind, temperature drops etc. Enforced automatic landing would be loss of the sensors.

Chapter 2.

A strong lack of primary literature regarding the aquatic chemistry.

How exactly was the data processed, which software and which parameters?

The DJI Phantom 4 pro multispectral image quality is not superior! How is the radiometric resolution?

Are images comparable? Was a orthomosaics created? Noise?

The authors present 28! Vegetation indices for multispectral analysis, without proper citation. However the camera features only 5 bands! Clearly there is a strong autocorrelation among the indices, and many could be omitted, also a dimensionality reduction such as PCA or MNF could be simply used.

It is unclear how and where the five modelling techniques were employed. How many image training samples?

Chapter 3.

This chapter is contains two long tables. Could be shown in a graph.

What could be physical or ecological reasons that the RNDVI performs best?

In figure 4 the unit is cells/ml? Is there any reference in the text to that? What does it mean? The word cell is not mentioned once.

Chapter 4.

Where there any secci/ view depth experiments?

The random forest is just 0.08 worse than CNN, or 1.6 µg/l…but quite straight forward to implement.

Chapter 5.

This manuscript tries to highlight CNNs, but how this procedure could be tested by other researchers remains unclear.

Reviewer 2 Report

Dear AUthors

The paper presented the used o UAV to evaluated the chl A in lake water with different indexes and different methods of analysis.

I have different main comments regarding this paper.

The method of measurements of the Chl a. in the lake is not described. I think you need to improve strongly this part to better understand the work.

You use different indices based on the UAV images. I think it could be great to add a PCA on the different indices to show the collinearity and complementarity.

I think that you could also reduce the number of decimals in the RMSE and R².

Reviewer 3 Report

The manuscript addresses the estimation of chlorophyll a in water from UAV multispectral data. Several improvements should be made before it could be considered for acceptance.

Comments and suggestions:

The title could be modified to "Water Chlorophyll a Estimation Using UAV-based Multispectral Data".

Lines 38-41: Add a reference.

Lines 45-48: Add a reference.

Lines 111-115: Add a reference.

When plural, "vegetation indices" should be used instead of "vegetation index". Please replace it throughout the manuscript.

The legends of most Figures and Tables are short and are not self-explanatory of what is described in the Figures.

For instance, in Figure 1 it could be "Study area overview and its location in the Erhai Lake (a), the used unmanned aerial vehicle and (b) location of the sampling points." Moreover, add (a) and (b) in the Figure.

No details regarding data alignment are provided, were the acquired imagery subjected to photogrammetric processing? Or the vegetation indices were directly computed from each image? Was any imagery alignment method employed? Please include all these information in the manuscript.

Please add the references for each vegetation index in Table 2.

Section 2.4 is irrelevant as it only describes the machine learning regressors and not its parameters. In Section 2 there is no mention to the univariate regression analysis

How the association with the sampling place and the data to be used by the UAV was made? Is the whole image used or only part of its pixels?

There is no information regarding the dataset splitting (training and testing) and the hyper parameterization of the evaluated machine learning techniques.

The discussion does not compare the obtained results with the results obtained in previous studies.

Please see other comments and suggestions in the attached PDF.

Reviewer 4 Report

1.     Chl-a is not an index. Please rewrite the first sentences (Line 13)

2.     Please rewrite Line 14.

3.     You collect the data for 2 months? Is it sufficient? Any other reference to support your data collection?

4.     Line 132 - DJI P4M and it’ multispectral – please add the company and country that produce the UAV and sensor. Please refer journal on how people write the correct way.

5.     Do you refer to any other sources for table 1? Add the citation.

6.     Do you conduct the calibration using a white panel before and after flying the drone? Please explain the method in detail.

7.     Line 214: To build and test the model, the total dataset was randomly divided into a training 214 set and a test set. – how come you randomly divided, normally we divided 70% and 30%. Please check the journal paper.

8.     Line 317 - superior – what do you mean with this word, use measurable words!

9.     You Need to rewrite many sentences in the text.

10.  Please check that the data is sufficient for this analysis, 2 months of the dataset for me is not enough. 

Round 2

Reviewer 1 Report

The authors tried their best to answer the raised question, but my main concern of a transparent and reproducible line of research has not been addressed.

Reviewer 3 Report

The authors improved the manuscript by addressing the majority of the comments and suggestions from the previous review.

Table 2 should include the reference to the studies where each vegetation index was origanily proposed. Moreover, there is a typo in "(NDVII)" replace by "(NDVI)" and consider to put the vegetation index acronyms in the calculation formula colunm (e.g: NDVI =  (??? − ?)/(??? + ?)).

In the legend of Table 4 replace "R2" by "Coeficient of determination (R2)".

In Section 4.4 the authors could write some statements about the possible use of UAV-based hyperspectral imagery as it would provide the possibility to compute more vegetation indices due to the higher spectral resolution, consider citing the following study:

Adão, T., Hruška, J., Pádua, L., Bessa, J., Peres, E., Morais, R., & Sousa, J. J. (2017). Hyperspectral imaging: A review on UAV-based sensors, data processing and applications for agriculture and forestry. Remote sensing, 9(11), 1110.

Reviewer 4 Report

Most of the previous comments in the answer are not include any academic references like journal papers.